# Chronic Hepatitis B: A Summarized Anecdote of Complexities in Natural History, Treatment, and Complications

Nicholas Noverati [1] , Jay W. Jun [2], Vivian Yan [2], Dina Halegoua-DeMarzio [1] and Hie-Won Hann [1,*]

1   Division of Gastroenterology and Hepatology, Department of Medicine, Thomas Jefferson University Hospital, Philadelphia, PA 19107, USA; nicholas.noverati@jefferson.edu (N.N.); dina.halegoua-demarzio@jefferson.edu (D.H.-D.)

2   Sidney Kimmel Medical College, Thomas Jefferson University, Philadelphia, PA 19107, USA; jay.jun@students.jefferson.edu (J.W.J.); vzy001@students.jefferson.edu (V.Y.)

*   Correspondence: hie-won.hann@jefferson.edu

**Abstract:** Chronic hepatitis B is still a disease process that affects millions around the world. Serologies used to diagnose and follow the progression (or resolution) of the disease can be confusing for clinicians. Further, throughout years of treatment, there may be nuances in presentation that complicate management even further. In this short communication, we highlight six themes in response to treatment and outcomes, including complications. We have the unique perspective of following many patients over extended periods of time at our institution, which has brought these themes to life in order that they can be shared with other clinicians who may encounter similar situations.

**Keywords:** chronic hepatitis B; host response; longitudinal observations

## 1. Introduction

Hepatitis B virus (HBV) can be controlled by the immune system in some patients with no resultant significant morbidity, but in others, the virus can lead to serious complications. Clinicians worldwide struggle with interpreting the different phases of infection and how to best treat the patient. Currently, most patients are committed to years of treatment with antiviral agents such as nucleos(t)ide analogs. Research is progressing in the field to find curative drugs that would ideally allow for more direct mechanisms to control the virus.

At our institution, we have had the unique experience of building a large, longitudinal cohort of patients with chronic hepatitis B (CHB). One of the many benefits of this is the continued ability to observe patterns over time, in order to better understand the virus and its interaction with its host. Largely, we have noticed not only themes of how the virus behaves over time and in response to treatment, but also how the host's response can be variable. The intermingling of the two is complex, and can be influenced by external and internal factors.

The purpose of this short communication is to summarize some of our observed themes in the natural history, treatment and complications of CHB. Sharing these experiences with the medical community will allow others to be better equipped with nuances that may arise in the management of similar patients in practice.

## 2. Vagaries in the Host Response to Viruses

Patients with CHB may be on treatment with, commonly, nucleos(t)ide analogs if HBV DNA is >20,000 IU/mL and ALT levels greater than two times the upper limit of normal for 3–6 months [1]. One might ask, if HBV DNA is elevated or fluctuates to extremes, without ALT elevation, do patients carry a higher risk of complications such as cirrhosis or hepatocellular carcinoma (HCC) that may call into question whether treatment is needed? We have observed two themes in patients not on current treatment or treatment naïve: one in which patients have persistently low levels of ALT and HBV DNA over time,

and on in which patients have low levels of ALT but fluctuating HBV DNA [2]. Even having a family history of complications of HBV, patients can still do quite well, as we have observed following them for over 30 years [2]. Further, even if DNA levels reach an extreme temporarily, patients still seem to do quite well without developing serious complications.

A lesson in our practice is that those with fluctuating levels of HBV DNA require closer follow up and monitoring of their transaminase levels and for any symptoms that could indicate a possible complication of their disease, but often do not need treatment and do quite well in the long term. Further, our observation stresses the reality that the host's response to HBV infection can be rather variable. Our data show that there is a "struggle" happening between the host immune system and viral replication over time, which is evident in extremes of laboratory values (high viral DNA and/or transaminase levels).

### 3. Family Clusters

In endemic regions, patients with CHB are commonly infected at birth, which leads to families with many offspring who are infected. It is thought that given the immune system has yet to fully mature at such a young age, patients are less likely to clear the virus completely at the time of infection, leading to a chronic infective course [3]. Given vertical transmission is common, especially in Asian countries, it has become of particular interest to study family clusters of infection to better understand the natural history of the disease.

It has been our overall observation that despite the assumed similar genetic makeup of virus and similar genetics within families, outcomes of CHB can still vary greatly. For example, in a family with identical twin sons who were infected at birth from their mother, one twin developed HCC, while the other remained asymptomatic and symptom-free for years, as shown in Figure 1 [4]. The twin with the less desirable outcome had greater levels of stress in his life, and may have had exposure to other environmental toxins, such as alcohol. Thus, the possibility of environmental factors making for an imbalance between the immune system's ability to "fight" the virus was presumed likely in this particular family instance. Seemingly, not only can different environmental factors have directly negative impacts on the liver and immune system, but an individual host's ability to manage such a stressor may also vary greatly. For example, one individual may be better equipped to cope with social stressors than another individual, with the latter suffering greater degrees of downstream effects of this.

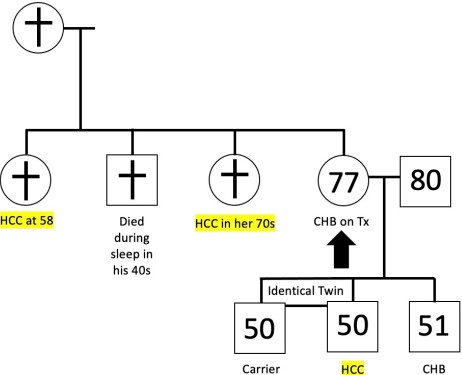

**Figure 1.** † = deceased; numbers correspond to age at time of presentation. CHB: chronic hepatitis B, Tx: treatment, HCC: hepatocellular carcinoma. The larger arrow indicates the propositus. All siblings of the 77-year-old mother were HBsAg (+). Highlighted color corresponds to patients with HCC.

Overall, the lesson from our practice and observation over many years has been that family history and the course of family members are not reliable predictors of how an individual may respond to chronic infection.

### 4. The Importance of an Individual's Response to Environmental Factors

Considering that some individuals with CHB have relatively uneventful courses of infection whereas others develop serious complications such as HCC and cirrhosis, one might ask why this is the case. The more obvious risk factors for a complicated course might include exposure to hepatotoxins including alcohol, drugs (both prescribed or illicit), co-infections with another virus, etc. However, an uncommonly thought of but certainly relevant risk factor is one that is more internal: a patient's ability to cope with external life stressors. We have observed over the years, anecdotally, many patients' stories echo a similar theme: when serious life events or circumstances cause stress in a patient's life, they tend to do worse. A specific example was published; it tells the story of a man with cirrhosis and CHB that developed an HCC tumor during a time of significant financial stress that then decreased in size when this stressor was relieved [5]. Though there may be other more complex factors that went into this, it is difficult to discredit the possibility that significant, chronic stress in someone's life can physically alter the immune system's regulatory mechanisms used to "fight" viruses or even cancer [6].

An important lesson learned in our longitudinal practice is that stress should be regarded as an important and potentially intervenable risk factor in patients with CHB.

### 5. A Functional Cure Is Possible

Nucleos(t)ide therapy remains the first line in the treatment of CHB. As the medical community awaits advancements in curative drugs, clinicians may wonder if a "functional" cure is possible with current therapies. A functional cure is defined by the loss of both HBV DNA and Hepatitis B surface antigen (HBsAg) [7–9]. In our practice, we have shown that when persistently treated with nucleos(t)ide analogues (and even with the use of several different agents), over a mean of 15.3 years, a functional cure can be achieved with no serious side effects from stopping therapy [10]. Though the rate of this happening may remain low in the general population of patients treated for HBV, it is still possible. Further, it can be helpful and reassuring for the clinician deciding to stop therapy when HBV DNA and HBsAg are negative that data support that patients will remain HBsAg-negative and "cured" of their infection [10]. The costs and side effects of medications should be considered, and clinicians should be actively following serologies in their patients to determine when they can potentially stop treatment to avoid negative outcomes.

### 6. HBeAg May Persist and Should Be Taken Seriously

Typically, in the natural history of chronic infection, seroconversion from hepatitis B e-antigen (HBeAg)-positive to -negative status usually signifies a patient no longer being infective and viral replication being very low. However, how may HBeAg positive status be interpreted in the setting of undetectable HBV DNA over decades of continued antiviral treatment?

At our institution, we have observed a small cohort (19) of HBsAg-positive patients with persistent HBeAg, despite negative HBV DNA for over a decade on antiviral therapy. All nineteen of these patients have done well clinically, without progression of liver disease or development of HCC. Given that typically, cccDNA should also be diminished when HBV circulating DNA is low from the antiviral effect, we hypothesized that persistent HBeAg may not be from cccDNA, but rather from integrated HBV DNA (pending confirmatory laboratory data). Though HBV DNA can be negative (>10 years), we have found it is important that these patients remain on antiviral therapy if HBeAg remains positive, even if HbsAg becomes negative, as we have seen that those who have had antivirals discontinued developed serious complications, such as the development of HCC.

### 7. Those with HBV-Associated HCC on Antiviral Therapy May Do Worse than Treatment-Naïve HCC Patients

The development of HBV-associated HCC is a complex pathogenic process with several different proposed theories, one of which is the role of HBV DNA integrating into a

host's genome [3,11]. It is known that even patients who are being actively treated with antivirals may have had some degree of integration of HBV DNA during the course of chronic hepatitis B occurring at low levels, which as a result, can play a role in oncogenesis over time.

In our practice, we have observed groups of patients who have been successfully treated with nucleos(t)ide analogs for several years (up to 20), who still eventually develop HCC [12,13]. When compared to another group of patients with CHB who are naïve to treatment, patients who have been on antivirals seem to do worse; their cancer seems to be more aggressive. Although it is not fully understood why this has been observed, some proposed reasons include the accumulation of unfinished viral products during the halting of viral DNA transcription, which may further promote integration of these products into host DNA [12,13].

A lesson from this observation is that the clinician who encounters HCC in a patient who has been successfully treated for HBV with antivirals for many years may want to be more proactive in referral for liver transplant, compared to the case of a patient who has not been exposed to antivirals.

## 8. Conclusions

Our observations offer a unique perspective as a result of following patients with chronic hepatitis B over many years. We have presented both nuanced findings and overarching themes that can help the clinician in managing similar patients in their practice.

In particular, we have seen that individuals can respond very differently to the virus, as evidenced by fluctuations in laboratory values (HBV DNA and ALT). Some of the reasons why someone may respond differently to the virus include their family history, as well as their exposure to and ability to cope with environmental stressors. Irrespective of how an individual responds to the virus, a functional cure may be possible if they remain on therapy long enough. Frequent monitoring of hepatitis B serologies can help determine when it is safe for a patient to stop therapy, and doing so may be important to avoid the side effects of the medication and reduce any possible cost burden to the patient.

Largely speaking, one uniting theme we have observed is that the struggle between host and virus has been evident over the years and stresses the importance of treatment decisions being made on an individual case-by-base basis. No one patient is created alike. Until curative drugs are available, treatment will remain largely variable.

**Author Contributions:** N.N., J.W.J., V.Y., D.H.-D. and H.-W.H., all contributed substantially to this manuscript. Conceptualization, N.N. and H.-W.H.; Writing—Original Draft Preparation, N.N., J.W.J. and V.Y.; Writing—Review and Editing, N.N., D.H.-D. and H.-W.H.; Supervision, H.-W.H. All authors have read and agreed to the published version of the manuscript.

**Funding:** This research received no external funding.

**Institutional Review Board Statement:** This study was submitted to our IRB and deemed EXEMPT from IRB review on 5 December 2022, pursuant to Title 45 Code of Federal Regulations Part 46.101(b) governing exempted protocol declarations.

**Informed Consent Statement:** Not applicable.

**Data Availability Statement:** The data presented in this study are available on request from the corresponding author. The data are not publicly available due to the communication style of this article.

**Conflicts of Interest:** Hie Won Hann serves the National Advisory Board of The Gilead Sciences and receives grant funding from Gilead and Assembly Biosciences. Dina Halegoua-DeMarzio is a consultant for Intercept, GlympseBio, Pfizer, and 89Bi, and receives research grant support from Intercept, BMS, Genfit, Novo Nordisk, Viking, Galmed, Pfizer, and Galactin.

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
