# Peer review of "Chronic Hepatitis B: A Summarized Anecdote of Complexities in Natural History, Treatment, and Complications"

_livers, doi:10.3390/livers4010003_

Round 1
Reviewer 1 Report
Comments and Suggestions for Authors
This manuscript is a series of anecdotes that helps demonstrate the wide range of experiences people living with chronic HBV infection can have and that clinicians need to be aware of.
The start of this manuscript is not auspicious:
Lines 22 – 24: “Hepatitis B virus (HBV) can be cleared by the immune system in some patients. For others, it becomes a chronic infection. For few, infection resolves but they remain at risk for re-activation in the future.”
We have to be precise in our language to avoid further confusing people.
Do you mean “controlled by the immune system”? I am not sure how we can definitively say a patient does not have any evidence of cccDNA in any hepatocytes. In this case, depending on the age of the infected person, few to most people develop immune control, but all of these people remain at risk for reactivation if they develop certain immunosuppressive conditions.
Lines 27 – 28: “Research is progressing in the field to find cure drugs, similar to what we now have available in the treatment of Hepatitis C.” This is not true. HCV does not have any nuclear-integrated genetic repository and can be completely eliminated from the host. The most likely goal for advanced HBV treatment will be development of immune control, similar to what most adults exposed to HBV have now. These people will likely still be at risk for HBV reactivation if they develop an immunosuppressive condition.
Lines 120 – 122: “However, how may HBeAg positive status be interpreted in the setting of undetectable HBV DNA over decades as well as HBsAg negativity?” The subsequent anecdote described a population of people with suppressed HBV DNA on antiviral medications. Were these people all HBsAg negative? If so, as per “5. Functional Cure is Possible”, do you test all patients who have cleared HBsAg on antiviral therapy for HBeAg, and if HBeAg remains positive, then you continue that group on antiviral therapy? This is not in any guidelines that I am aware of and if data support this approach, it would be worth showing the actual data.
Reviewer 2 Report
Comments and Suggestions for Authors
General comment
This short communication is unusual for a scientific journal like livers. It does not provide accurate data or description of cases but views and comments on the consensus concerning chronic HBV infections. The paper is a kind of auto-review of 7 case reports (“anecdotes”) published by the authors previously: refs. 2, 4, 5, 6, 10, 12, 13.
For the non-expert in HBV infection, it may provide new insights, but it does not provide clear advice how to treat patients living with HBV.
Specific points
1. Figure 1 is interesting but insufficiently explained. A more explicit legend is urgently required.
a. These are obviously descendants of a HBsAg positive mother. Were all HBsAg positive?
b. Was the person dying at 40 years an inapparent HBV carrier?
c. What does CHB on Tx mean? Is a transplantation at the age of 77 normal?
d. Please replace HBsAb by the recommended nomenclature antiHBs.
2. Ref. 2, 5, 12, 13 are incomplete. The journal is missing.
3. L118-130. This para. should be modified. HBeAg is probably not expressed from integrated HBV DNA, because the integration interrupts the circular HBV DNA genome leading to pregenome and preC mRNAs. HBeAg is expressed from the episomal cccDNA and indicates transcriptional activity of the cccDNA including expression of pgRNA. HBsAg may be negative because of diagnostical escape mutation, an efficient antiHBs response or too low levels. Transcription of preC RNA competes with transcription of pgRNA and may reduce replication of HBV DNA to undetectable levels. Furthermore, antiviral therapy suppresses reverse transcription of pgRNA but not translation of preC mRNA. Thus, the thoughts of the authors are partly correct, but the connection of HBeAg to the development of HCC is not clear.
Comments on the Quality of English LanguageThe English is ok. But the scientific style is unusual. This short communication is for a scientific journal like livers. However, it does not provide accurate data or description of cases but views and comments on the consensus concerning chronic HBV infections. The paper is a kind of auto-review of 7 case reports (“anecdotes”) published by the authors previously: refs. 2, 4, 5, 6, 10, 12, 13.
Reviewer 3 Report
Comments and Suggestions for Authors
This is a narrative statement on some aspect of therapy in patients with chronic HBV infection. The authors refer to their center experience to comment on some findings.
I have a few questions:
Point2: Are the author refer to patients without current treatment only? These should be monitored either way (especially if not therapy is inititated)? I do not understand the take-home message.
Point 3/4: Those points discuss similar aspects->that environmental aspects influence disease course. Some more data woukd be interesting.
Fig.1 : legend refers to serological parameters, but where are they?
Point 7: Very interesting, is there any substantial data? But also, most patients with chronic HBV infection and HCC should be evaluated for LTx anyway.
Round 2
Reviewer 3 Report
Comments and Suggestions for Authors
The authors have revised the manuscript.
However, in Fig. 1 i still cannot find HBsAG/antiHBs-status indicated with +/- as the legend suggests. I understand, that all siblings from index patient (f 77y) were HBsAG+, so maybe just remove the sentence "+/- indicates HBsAG and antiHBS status, respectively"?
Comments on the Quality of English Languagemostyl fine.
Author Response
Thank you for your comment. This has now been resolved; the sentence in the figure legend has been deleted.